# Development of a Novel Method for the Clinical Visualization and Rapid Identification of Multidrug-Resistant *Candida auris*

X. R. Zhang,[a,b] T. Ma,[a] Y. C. Wang,[a] S. Hu,[a] Y. Yang[a]

[a]Bioinformatics Center of AMMS, Beijing Key Laboratory of New Molecular Diagnosis Technologies for Infectious Diseases, Beijing Institute of Microbiology and Epidemiology, Beijing, People's Republic of China
[b]School of Life Sciences, Hebei University, Baoding, People's Republic of China

X. R. Zhang, T. Ma, and Y. C. Wang contributed equally to this article. The order was determined by the corresponding author after negotiation.

**ABSTRACT**   Outbreaks of multidrug-resistant *Candida auris* infections, associated with a mortality rate of 30% to 60%, are of serious global concern. *Candida auris* demonstrates high transmission rates in hospital settings; however, its rapid and accurate identification using currently available clinical identification techniques is challenging. In this study, we developed a rapid and effective method for detecting *C. auris* based on recombinase-aided amplification combined with lateral flow strips (RAA-LFS). We also screened the appropriate reaction conditions. Furthermore, we investigated the specificity and sensitivity of the detection system and its ability to distinguish other fungal strains. *Candida auris* was accurately identified and differentiated from related species at 37°C within 15 min. The minimum detection limit was 1 CFU (or 10 fg/reaction) and was not affected by high concentrations of related species or host DNA. The simple and cost-efficient detection method established in this study exhibited high specificity and sensitivity and successfully detected *C. auris* in simulated clinical samples. Compared with other traditional detection methods, this method greatly reduces the time and cost of testing and is thus suitable for hospitals or clinics in remote underfunded areas for screening *C. auris* infection and colonization.

**IMPORTANCE**   *Candida auris* is a highly lethal, multidrug-resistant, invasive fungus. However, conventional methods of *C. auris* identification are time-consuming and laborious and have low sensitivity and high error rates. In this study, a new molecular diagnostic method based on recombinase-aided amplification combined with lateral flow strips (RAA-LFS) was developed, and accurate results could be obtained by catalyzing the reaction at body temperature for 15 min. This method can be used for rapid clinical detection of *C. auris*, consequently saving valuable treatment time for patients.

**KEYWORDS**   *Candida auris*, lateral flow strip, rapid detection, recombinase-aided isothermal amplification assay

Invasive fungal infections (IFIs) result in 1.6 million fatalities annually, a number equivalent to the total annual deaths from malaria and tuberculosis (1). On 25 October 2022, the World Health Organization (WHO) published the first List of Priority Fungal Pathogens (FPPL) for invasive fungal diseases (IFDs). In this list, the WHO enumerates a total of 19 deadly fungi divided into three priorities: extremely, highly, and moderately important. In the extremely important group, the WHO lists four fungi including *Cryptococcus neoformans*, *Candida auris*, *Aspergillus fumigatus*, and *Candida albicans*. *Candida auris* is a recently discovered emerging pathogenic fungal species that has garnered widespread attention due to its multidrug resistance and increased infection outbreaks (2). *Candida auris* was first isolated in Japan in 2009 from the external auditory canal of a patient (3). Since then, the rapid and simultaneous global emergence of this pathogenic fungus has been described

Address correspondence to Y. Yang, y_ying_77@163.com.
The authors declare no conflict of interest.

across all continents, except for Antarctica. *Candida auris* is separated into four geographic clades which differ by >10,000 single nucleotide polymorphisms (SNPs), namely, the South Asian (clade I), East Asian (clade II), African (clade III), and South American (clade IV) clades (4). Recently, a fifth clade has been reported from Iran (5, 6). The high number of SNPs among clades suggests the emergence of *C. auris* in multiple locations simultaneously, as opposed to a clonal source. *Candida auris* infections have predominantly been associated with health care. In the context of nosocomial candidemia, *C. auris* is overrepresented and is endemic to South Africa and India, accounting for 15% and 5% to 30% of the nationally reported candidemia cases, respectively (7). Nosocomial spread causing protracted outbreaks involving critical and intensive care unit (ICU) settings and immunocompromised cohorts has been reported in Europe (Spain and the United Kingdom), the United States, and Venezuela. Other countries, such as Norway, Germany, and Australia, have reported sporadic cases (8). *Candida auris* infections have been documented in multiple (>40) countries worldwide, affecting between 5% and 10% of *C. auris*-colonized patients (9). Among patients colonized with *C. auris*, 42.9% reportedly test positive in the nostrils, 40.4% on palms and fingertips, and 35.7% on toe webs (10). The clinical spectrum of *C. auris*-related infections ranges from mild, superficial infections (such as otitis media) to invasive candidiasis due to other species (11). The epidemiology for candidemia is similar to that for other *Candida* species. At-risk groups include those at the extremes of age, ICU patients, and patients with underlying immunosuppression or chronic diseases, particularly following exposure to health care settings. Crude mortality rates of *C. auris* invasive fungal disease remain high (30% to 60%) (12). Rapid and specific species identification is necessary for accurate treatment, improved prognosis, and control of multidrug-resistant fungi (9, 11, 12). However, conventional methods of fungal identification, including culture, microscopic examination, and physiological as well as biochemical examination, are time-consuming and laborious to use, and they also have low sensitivity and high error rates (13, 14). Furthermore, the most commonly used clinical microbiological identification systems, such as the Vitek 2, API-20C AUX, and BD yeast identification systems, often misdiagnose *C. auris* as other closely related species, such as *Candida haemulonii*, *Candida duobushaemulonii*, *Candida pseudohaemulonii*, *Candida guilliermondii*, and *Candida parapsilosis* (15), resulting in high missed and false diagnosis rates in patients with *C. auris* infection (16). Moreover, specific diagnostic methods, such as metagenomic next-generation sequencing (mNGS), quantitative PCR (qPCR), and matrix-assisted laser desorption ionization–time of flight mass spectrometry (MALDI-TOF MS), usually require additional expensive equipment and trained technicians. These factors are associated with high running costs, thus posing limitations to numerous medical institutions (17–19). The high costs and low accuracy of the existing diagnostic tools might lead to underestimations of the prevalence of *C. auris* globally, especially in regions with limited health care resources (such as Africa and Southeast Asia) (20): all known outbreaks and infection cases originate from developed countries, such as the United States, United Kingdom, and Germany, and cities with advanced medical facilities in low-income areas (such as New Delhi, India), further corroborating this conjecture (16, 21, 22).

The risk of death reportedly increases by approximately 50% for each day of delay in administering an effective antifungal therapy. Therefore, it is crucial to correctly identify the pathogenic fungi at an early stage and quickly initiate appropriate systemic antifungal therapy (23). However, in remote and resource-limited areas, there is generally a lack of proper diagnostic facilities, equipment, and well-trained staff. Consequently, an effective diagnostic technology should meet the following requirements: low cost, high sensitivity, specific diagnosis, ease of operation, speed, and high portability. The emerging thermostatic detection technology of recombinase-aided amplification combined with lateral flow strips (RAA-LFS) meets these criteria and is thus the best choice for screening super-resistant invasive fungi in remote and resource-limited areas (24–26).

In this study, a detection method combining recombinase-aided amplification and rapid lateral flow strips (RAA-LFS) was developed. It accurately identifies and efficiently distinguishes *C. auris* from other related species within 15 min at 37°C. This system was

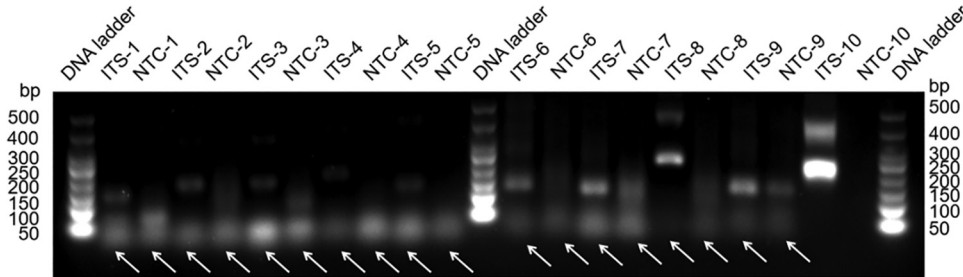

**FIG 1** Screening of primer pairs according to their RAA performance. Agarose gel image analysis illustrates the amplicons of primer pairs targeting the ITS gene. Each primer pair is labeled above each lane. The DNA ladder band size is indicated. Primer dimers are indicated by white arrows. NTC, no-template control (numbered for each respective primer pair).

further verified using standard strains and simulated clinical samples, thereby providing intelligible and easily interpretable results.

## RESULTS

**Design and screening of the RAA-LFS detection system primer-probe set.** Based on the internal transcribed spacer (ITS) sequence from the *C. auris* genome (NCBI reference sequence NR_154998.1), we designed 10 candidate primer pairs using the Primer Premier 5.0 software (see Table S2 in the supplemental material). We established a control group without a template and excluded pairs with evident primer dimer formation. We then selected the primer pair ITS-10, which demonstrated the highest amplification efficiency (Fig. 1). The RAA amplicons of 10 primer pairs were verified using Sanger sequencing, confirming 100% identity with the *C. auris* ITS gene sequence (NR_154998.1). Furthermore, we obtained the candidate probe P by extending the 3′ end of the upstream primer F10 by 16 bp. The original downstream primer sequence remained unaltered, with only a biotin modification set to R10′ at the 5′ end. We also used the same primer design software to predict potential dimer formation between the candidate probe sequence and the original downstream primer sequence. Accordingly, we redesigned four primers upstream of the candidate probe sequence and used them with the candidate probes and the downstream primers for the RAA-LFS test. We selected the pairs that demonstrated the highest amplification efficiency and no false-positive results, specifically the F10′-1/R10′/P combination (Fig. 2). Notably, we observed that the F10′-1/R10′/P primer-probe combination efficiently detected *C. auris* and distinguished it from *C. pseudohaemulonii*, *C. duobushaemulonii*, and *C. haemulonii* (Fig. 3). These three *Candida* species are phylogenetically closest to *C. auris*, as indicated by the phylogenetic tree of the whole genome of *C. auris* and other common *Candida* species (Fig. S3), as well as by ITS sequence alignment.

**Optimization of RAA-LFS detection conditions.** We optimized the reaction conditions for the RAA-LFS system using both temperature and time gradients. The temperature ranged from 35 to 45°C (increments of 2°C), and the reaction times ranged from 5 to 35 min (increments of 5 min). We analyzed the results by observing the intensity of the test line on the LFS. We observed that the pink color did not appear on the test line for reactions conducted at 45°C; this could be attributed to temperature-induced enzyme inactivation. However, the pink color appeared on the strip for reactions conducted at 39°C (10 min). Additionally, the color gradually intensified after 15 min for reactions carried out at 37°C and 39°C. After 20 min (until 35 min), the band color remained unchanged (Fig. S4). Therefore, considering the reaction conditions that are easiest to implement in practical applications, we selected 37°C and 15 min for subsequent RAA-LFS detection.

**Specificity of the RAA-LFS system.** To confirm the selectivity and specificity of the primer-probe set, we performed RAA-LFS amplification tests using 12 reference strains of *C. auris*, 4 strains of *C. haemulonii*, 3 strains of *C. pseudohaemulonii*, and 4 strains of *C. duobushaemulonii*. We also included other closely related species, such as *Candida rugosa*, *C. albicans*, *C. neoformans*, *C. parapsilosis*, *Candida glabrata*, *C. guilliermondii*, *Candida tropicalis*, *Candida krusei*, and *Candida dubliniensis*, and common pathogenic bacteria

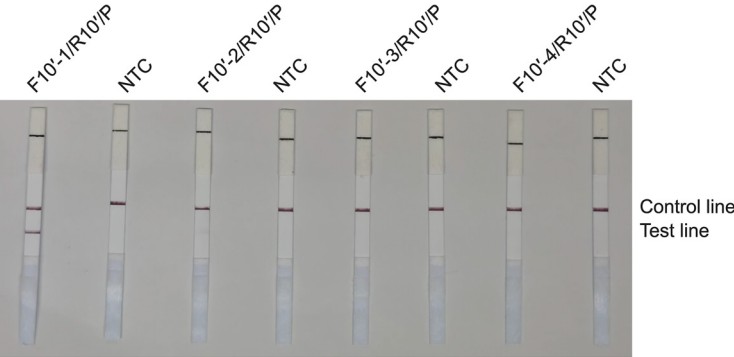

**FIG 2** RAA-LFS test results using ITS-10′ primer-probe sets. Lateral flow strip (LFS) results of recombinase-mediated isothermal nucleic acid amplification (RAA) using different primer-probe sets. The *Candida auris* genomic DNA was used as the template, and reactions were performed at 39℃ for 15 min. The name of each primer-probe set is indicated at the top of each strip. NTC, no-template control. The positions of test and control lines are marked on the right.

(*Escherichia coli*, *Staphylococcus aureus*, *Enterococcus faecalis*, and *Klebsiella pneumoniae*). We discovered that our RAA-LFS system positively detected the 12 reference strains of *C. auris* (Fig. 4), whereas negative results were observed when other closely related fungi and pathogenic bacteria were tested (Fig. 5A and B). This confirmed the accuracy and precision of the RAA-LFS system for *C. auris*, and the results indicate that it is unlikely to cross-react with other pathogenic bacteria and fungi. Furthermore, we observed that all test results for *C. haemulonii*, *C. pseudohaemulonii*, and *C. duobushaemulonii*, which are the *Candida* species phylogenetically closest to *C. auris*, were negative (Fig. 5A). Consequently, the RAA-LFS system detected *C. auris* and accurately distinguished it from *C. haemulonii*, *C. pseudohaemulonii*, and *C. duobushaemulonii*.

**RAA-LFS system detection limits for *C. auris*.** We subsequently tested 10-fold gradient dilutions of *C. auris* genomic DNA and cell suspensions to evaluate the sensitivity of the RAA-LFS system. Specifically, we employed a concentration gradient of genomic DNA ranging from $10^7$ to $10^0$ fg/reaction. Additionally, we used a concentration gradient of cell suspension ranging from $10^5$ to $10^0$ CFU/$\mu$L (2 $\mu$L *C. auris* genomic DNA/reaction). We

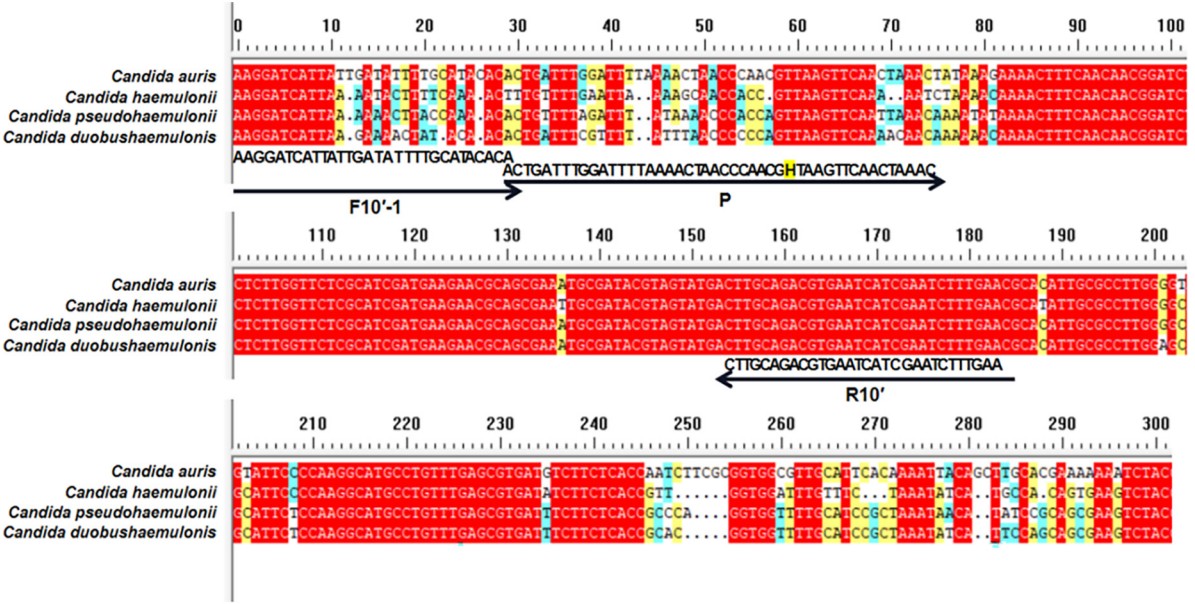

**FIG 3** Fragments targeted by various primer-probe combinations. The ITS sequence fragments of three closely related strains, *Candida pseudohaemulonii*, *C. duobushaemulonii*, and *C. haemulonii*, were compared with the ITS sequence of *C. auris*. Primer and probe sequences are indicated under the aligned ITS sequences. Arrow lines indicate the extension direction of primers and probes. Tetrahydrofuran (THF) sites are indicated by the letter "H."

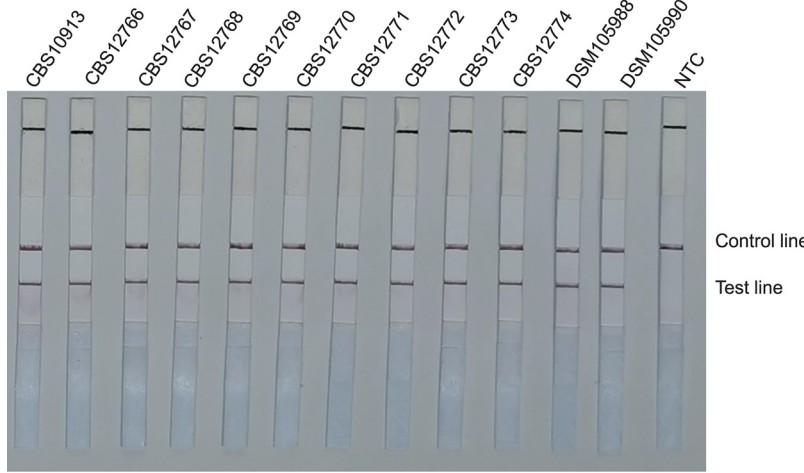

**FIG 4** Detection of standard reference strains of *Candida auris*. LFS results of RAA of different genomic DNA templates. Strain names are indicated on top of each strip. NTC, no-template control. The positions of control and test lines are indicated on the right of the image. All reactions were performed at 37°C for 15 min.

observed the pink band on the test line at $10^0$ CFU/$\mu$L of cell suspension. Intriguingly, the color gradually darkened with the increasing concentration of *C. auris* suspension (Fig. S5A). Similarly, we observed that the RAA-LFS system detected as low as 10 fg/reaction of *C. auris* genomic DNA (Fig. S5B).

Subsequently, we added $10^5$ CFU/$\mu$L, or 1 ng/$\mu$L, of *C. albicans* genomic DNA to either *C. auris* culture ($10^5$ to $10^0$ CFU/$\mu$L) or the genomic DNA ($10^7$ to $10^0$ fg/reaction) to evaluate whether contamination by other strains interferes with the sensitivity of the detection method. We observed that neither the presence of a high concentration of *C. albicans* cells (Fig. S6A) nor the genomic DNA (Fig. S6B) affected the detection limit of the RAA-LFS.

Consequently, the detection limit of the RAA-LFS system for *C. auris* was determined to be 1 CFU/reaction or 10 fg genomic DNA/reaction. The presence of other fungal DNA did not affect the sensitivity of the detection system. The RAA-LFS test results of the microdissected single cells and 10 cells demonstrated that the detection system could indeed detect 1 CFU (Fig. S7). Utilizing the grayscale value of the test strip as an index, the repeated experimental results of the sensitivity experiment confirmed that, upon reducing the sample concentration to $10^2$ CFU/$\mu$L and $10^3$ fg/reaction, the brightness of the test line band decreased significantly (Fig. S8).

**Application of the RAA-LFS detection technique for examining simulated clinical samples.** We subsequently evaluated the application of the RAA-LFS system in simulated clinical blood samples. To this end, we mixed different concentrations of *C. auris* cultures with human blood. We discovered that the RAA-LFS assay is capable of detecting *C. auris* in human blood samples with the same sensitivity observed in cultured cells (Fig. 6A). To determine the influence of human DNA on *C. auris* detection, we then mixed the genomic DNA of *C. auris* with DNA extracted from human blood at a 1:10 ratio. We observed that the human DNA did not significantly inhibit the assay, and it did not alter its detection limits (Fig. 6B). During grayscale analysis of the test line band, the band on the blood background was found to be less gray than the test band of the pure culture, indicating that the blood retained many impurities when using the Chelex-100 rapid DNA extraction method; however, this ultimately did not affect test results (Fig. S8). Simultaneously, to evaluate the use of the RAA-LFS system in screening patients colonized with *C. auris*, we prepared nasal swabs and urine samples containing different amounts of *C. auris*. The sensitivity of the RAA-LFS assay for both samples was consistent with that of pure cultured samples (Fig. S9). During grayscale analysis of the test line band, the brightness of the bands of simulated nasal swab samples and urine samples was not different from that of pure cultures (Fig. S8), indicating that the RAA-LFS method is suitable for the regular screening of patients colonized with *C. auris*.

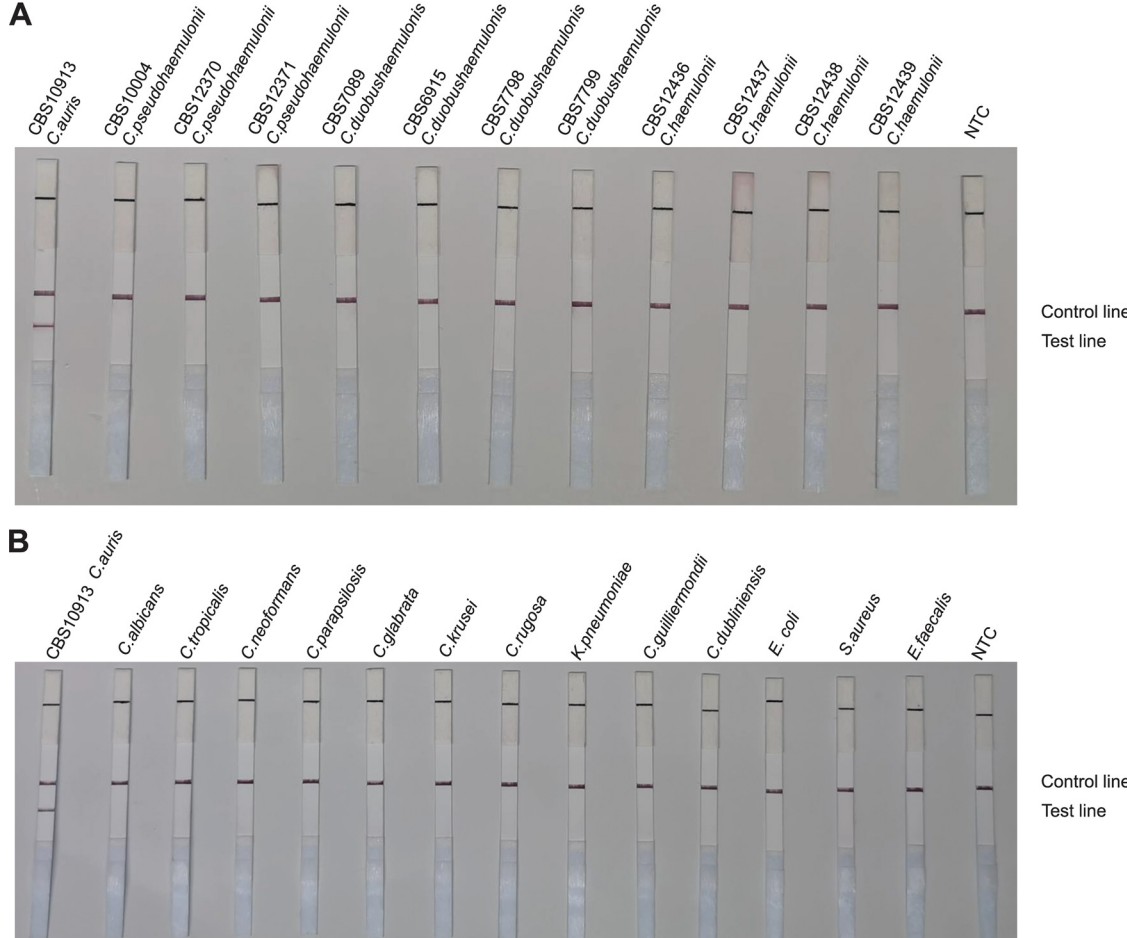

**FIG 5** Differentiation among common pathogens. Results of RAA-LFS detection of *Candida haemulonii*, *C. pseudohaemulonii*, and *C. duobushaemulonii* strains (A) and other related species and common pathogenic bacteria (B). The strains are indicated at the top of each strip. NTC, no-template control. The positions of control and test lines are indicated on the right of the image. All reactions were performed at 37°C for 15 min.

**Evaluation of RAA-LFS for the detection of *C. auris*.** The RAA-LFS system was used to detect *C. auris* in 48 simulated clinical blood samples. Its performance was compared with that of a qPCR system, which successfully detected 19 positive samples. The RAA-LFS detected 18 positive samples, whereas a false-negative result was given for 1 sample (Table S3), which was attributed to the quality control of the LFS. The kappa value was 0.956, the *P* value was <0.001, positive percent agreement was 100%, and negative percent agreement was 96.7%, which all indicate that the diagnostic results of the two methods are consistent (Table 1). Similarly, results of nasal swab and urine samples using the RAA-LFS and qPCR systems were consistent. For nasal swab samples, both methods detected 19 positive samples and 29 negative results (Table S4). For urine samples, both methods detected 21 positive samples and 27 negative samples (Table S5). No false-positive results were observed for either assay.

## DISCUSSION

The emerging multidrug-resistant fungal pathogen *C. auris* has become a global public health threat due to challenges in diagnosing and treating infections and its associated high mortality rate (27). The reasons for the emergence of multidrug-resistant fungal pathogens remain unknown (3). Outbreaks of *C. auris* have occurred in several hospitals worldwide in recent years (28, 29). Pneumonia has gradually attracted researchers' attention because of the COVID-19 pandemic, and *Candida* is also an important cause of pneumonia. COVID-19 complicated with *C. auris* infection is a more serious disease, particularly because *C. auris* readily

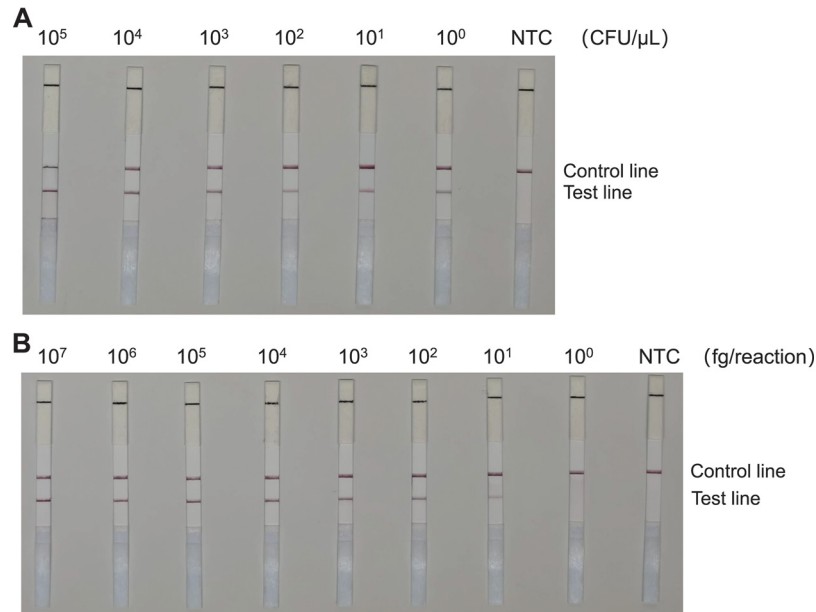

**FIG 6** Testing the RAA-LFS system using human blood samples. (A) Detection limits of the RAA-LFS system for *Candida auris* mixed into human blood samples. (B) The detection limit of *C. auris* DNA by the RAA-LFS system after mixing with human blood DNA at a 1:10 ratio. NTC, no-template control. The positions of the control and test lines are indicated on the right of the image. All reactions were performed at 37°C for 15 min.

causes nosocomial infections. In many cases, infections caused by *C. auris* are not recognized initially, leading to incorrect medication and delayed treatment (30–34).

Candidemia is an emerging invasive fungal disease. Traditional blood culture methods are considered the gold standard for its diagnosis, but conventional blood culture techniques are insensitive (approximately 50%) and time-consuming. Although many automated biochemical tests, such as the API Candida system (bioMérieux) and the Vitek 2 YST identifier (ID) card (bioMérieux), have reduced the turnaround time from more than 48 h to 15 to 24 h, these methods may not be accurate for some extremely closely related species of *Candida* (35).

Serum beta-D-glucan (BDG) determination plays an important role in diagnosing candidemia among critically ill patients admitted to the ICU. However, BDG levels measured may be lower in the case of infections caused by some non-*albicans* species, such as *C. parapsilosis* and *C. auris*. Furthermore, the overall sensitivity of BDG for diagnosing candidemia is low (47%, 95% confidence interval [CI] of 39% to 55%) (36), and false-positive results have been reported due to possible contamination with human immunoglobulins, gauze-containing dextrans, or cellulose membranes during hemodialysis. Alternatively, MALDI-TOF MS can identify closely related species and reveal the genetic and evolutionary relationships between and within species. The advantages of this test are the shorter turnaround time required for *Candida* identification, which is approximately 90 min for specimen processing after obtaining a fungus-positive blood culture, and the ability to detect rare *Candida* species. The limitations of this method include its high setup cost, the requirement of a useful

**TABLE 1** Determination of the coincidence rate between the RAA-LFS assay and qPCR analysis for the simulated clinical blood samples[a]

| | RAA-LFS assay | | |
|---|---|---|---|
| qPCR | Positive | Negative | Total |
| Positive | 18 | 1 | 19 |
| Negative | 0 | 29 | 29 |
| Total | 18 | 30 | 48 |

[a]Statistical values: kappa, 0.956; *P*, <0.001; positive percent agreement, 100; negative percent agreement, 96.7.

database, low sensitivity for direct testing of whole-blood samples (due to protein contamination), and potential misidentification of samples from polyfungal bloodstream infections (35). The costs associated with MALDI-TOF MS are high, and only certain large hospitals currently have access to the required equipment. Other diagnostic techniques, such as PCR, metagenomics next-generation sequencing (mNGS), and nanopore sequencing, require the efficient fragmentation of fungal cell walls and are difficult to operate (12). In addition, the results of mNGS and nanopore sequencing require analysis by bioinformatics professionals, making it challenging for clinical microbiologists to interpret the raw data (37, 38). In the 46 countries where cases of *C. auris* infections have been reported, many of the infecting strains have been identified through retrospective studies and reidentification experiments (39). *Candida auris* is easily misidentified; therefore, its spread may occur in countries that lack testing facilities and capacities. Despite the reduced number of cases of *C. auris* infection being reported, we believe that the real number is underreported (40). Therefore, there is an urgent need to develop a detection technology suitable for the diagnosis and screening of *C. auris* in economically disadvantaged hospitals in remote areas. Numerous molecular detection techniques have emerged in recent years. Compared to other molecular detection techniques, such as mNGS, MALDI-TOF MS, and qPCR, RAA-LFS offers advantages of high sensitivity and specificity, and it does not require expensive instrumentation and specialized staff. The LFS used in the RAA-LFS detection system is more affordable and convenient to use than other detection techniques, making the test suitable for remote and resource-limited areas (41). In addition to these advantages, the reaction time is also considerably reduced. The RAA-LFS detection system takes only 30 min from sample preparation to obtaining the results, saving valuable time for the treatment of critically ill patients while providing accurate results (42, 43).

RAA-LFS has been used to test various pathogenic microorganisms, including *Vibrio cholerae* (44) and *Helicobacter pylori* (45), as well as viruses such as African swine fever virus (24) and dengue virus (46). In this study, we employed RAA-LFS for the clinical visualization and detection of *C. auris* for the first time. This assay eliminates the need for any fluorescence detection equipment or electrophoresis devices, requiring only a thermostatic water bath to maintain the reaction temperature for effective reaction amplification. Furthermore, the method used in this study clearly distinguishes *C. auris* from its close relatives *C. haemulonii*, *C. pseudohaemulonii*, and *C. duobushaemulonii*. *Candida auris* is often misdiagnosed by commonly used clinical microbiological identification systems. The lower limit of detection for this system is 1 CFU or 10 fg genomic DNA per reaction (50 $\mu$L). In this study, we also created a complex detection background by mixing all the strains in the laboratory. Consequently, our detection system maintained a sensitivity of 1 CFU. To verify whether our detection system could accurately identify 1 CFU of *C. auris*, microdissection technology was used to isolate a single *C. auris* cell and detect it. Our results demonstrate that our detection system can truly detect 1 cell of *C. auris*. While evaluating the optimum reaction temperature and incubation time, we confirmed that the RAA core enzyme was active from 37 to 41°C. Neither high nor low temperatures were conducive to the reaction, and only 15 min was needed to achieve effective amplification. During the experiments, the product obtained in the RAA reaction was diluted 6-fold to avoid false-negative results caused by high product concentration in the test strip. Additionally, extraction and purification of DNA can be performed directly from clinical samples using the simple Chelex-100 boiling method. This method significantly reduces the detection time compared to the complex process of universal genomic DNA kit extraction methods. Likewise, this study evaluated the RAA-LFS detection system in blood and urine samples with complex backgrounds and in nasal swab samples from the most common colonization sites, and the results demonstrated that this system is suitable for various clinical samples without modifying the test sensitivity.

**Conclusions.** In this study, we developed a new molecular diagnostic method based on recombinase-aided amplification with lateral flow strips (RAA-LFS) for the rapid clinical detection of *C. auris*. Our method enables visual detection after RAA-LFS incubation at 37°C for 15 min. This straightforward approach effectively distinguishes closely related *Candida*

species, exhibiting excellent specificity and high sensitivity. Furthermore, it does not necessitate expensive equipment or specialized personnel, making it suitable for use in remote and underequipped hospitals to meet diagnostic needs. Clinical test results can be validated, and appropriate antifungal treatment can be promptly initiated. Consequently, our diagnostic tool holds significant value for the extensive and rapid screening of *C. auris*.

## MATERIALS AND METHODS

**Fungal strain culture conditions.** All strains were stored in the laboratory at −20℃ using the filter paper preservation method. All fungal strains were grown on Sabouraud dextrose agar (SDA) solid medium and cultured in an incubator at 25℃ for 48 to 72 h. Single colonies were selected and cultured in SDA liquid medium at 25℃ for 24 to 48 h. All bacterial strains were grown in LB solid medium and incubated at 37℃ for 12 to 18 h. Subsequently, single colonies were selected and cultured in LB liquid medium at 37℃ for 12 to 18 h. The cell density of the cultures was calculated by counting single cells in a cell counting plate under a microscope (model DM750; Leica Microsystems, Wetzlar, Germany). The source of all strains used in the experiment is shown in Table S1 in the supplemental material.

**DNA extraction.** For DNA extraction, 1-$\mu$L samples were mixed with 10% Chelex-100 (Fount Beijing Bio-Tech Co., Ltd., Beijing, China) and boiled at 100℃ for 10 min. Nasal swab samples were soaked in 10% Chelex-100 for 1 min and boiled at 100℃ for 10 min. Following centrifugation (4,000 × $g$ for 5 min), the purified genomic DNA obtained from the supernatant was used as the template [47]. From blood samples, genomic DNA was extracted by mixing DNA extraction phenol reagent (Beijing Solarbio Science & Technology Co., Ltd., Beijing, China) and the supernatant at a ratio of 1:1 after boiling. Unless specified, 1 $\mu$L of the heat-treated culture ($10^5$ CFU/mL) was used as the template. Isolated DNA was quantified using a Qubit 2 fluorometer (Thermo Fisher Scientific, Waltham, MA, USA) [48].

**Primers and probes.** The sequence of *C. auris* ribosomal DNA (rDNA) was sourced from the NCBI GenBank database (http://www.ncbi.nlm.nih.gov). The conserved ITS sequences of the *C. auris* rDNA ITS1 and ITS2 genes (NCBI reference sequence NR_154998.1) were used as the target sequences. Primer Premier 5.0 software (Premier Biosoft International, San Francisco, CA, USA) was employed for primer design. The NCBI BLAST function (http://blast.ncbi.nlm.nih.gov/Blast.cgi) was used for designing primers and probes. The top 10 candidate primer pairs were selected, and the upstream primer extending 16 bp to the 3′ end was used as a new candidate probe. The primer pairs and probe used were synthesized by Tsingke Biotechnology Co., Ltd. (Beijing, China).

The principles of primer design included the following: (i) primer length between 30 and 35 nucleotides (nt), (ii) amplicon length of approximately 110 to 140 bp, and (iii) exclusion of palindromic structures, continuous single-base repeat sequences, and internal secondary structure regions. Based on these principles, we used the following software settings: length, 30 to 35 nt; CG content, 20% to 70%; melting temperature ($T_m$) value, 50 to 100℃; maximum single-base repeat number, 5.

The principles of probe design included the following: (i) no overlap with the primer recognition site and a length of 46 nt; (ii) replacement of the 31st base by tetrahydrofuran (THF) residues; (iii) exclusion of palindromic sequences, internal secondary structure, and continuously repeated bases; and (iv) antigen labeling at the 5′ end (e.g., 6-carboxyfluorescein [FAM] and fluorescein isothiocyanate [FITC]) and blocking of the 3′ end with a C3-spacer. Additionally, we included antigen labeling at the 5′ end of the downstream primer (e.g., biotin).

**RAA procedures and electrophoresis.** The RAA technique uses recombinases, single-stranded binding proteins, and DNA polymerases to amplify a large number of target genes. Briefly, bacterium- or fungus-derived recombinant enzymes bind tightly to primer DNA at 25℃ to form a recombinase/primer complex, which acts on the DNA double-stranded nucleic acid template. The recombinase opens the double strand, whereas the single-stranded binding protein binds to the single strands, maintaining the double-stranded template in an open-stranded state. The recombinase/primer complex identifies an exact complementary sequence match, and it disassembles. The DNA polymerase binds to the 3′ end of the primer to initiate the synthesis of the new strand. The newly synthesized strand is subsequently used as a template, and the final amplification product, based on the template, is formed (Fig. S1).

The RAA reaction was performed using the basic RAA nucleic acid amplification reagent (Hangzhou Zhongce Bio-Sci & Tech Co., Ltd., Guangzhou, China) according to the manufacturer's instructions. Briefly, 50 $\mu$L of reaction solution consisted of 6.5 mg reaction dry powder, 25 $\mu$L A buffer (20% polyethylene glycol [PEG]), 2 $\mu$L forward primer (10 $\mu$M), 2 $\mu$L reverse primer (10 $\mu$M), 13.5 $\mu$L distilled water, and 5 $\mu$L of the DNA template. Subsequently, 2.5 $\mu$L B buffer (280 mM magnesium acetate [Mg(OAc)$_2$]) was added to the reaction mixture. After instantaneous centrifugation (4,000 × $g$ for 5 s), the reaction mixture was incubated at 39℃ for 30 min. At the end of the reaction, 50 $\mu$L of DNA extraction phenol reagent (Beijing Solarbio Science & Technology Co., Ltd.) was added. After mixing the sample well, the amplified products were purified using centrifugation at 12,000 × $g$ for 5 min. The sample within the supernatant was separated via electrophoresis using a 2.5% agarose gel.

**RAA-LFS detection with primers and probes.** The reverse primers and probe were modified with biotin and carboxyl fluorescein (FAM) at their 5′ end (Tsingke Biotechnology Co., Ltd., Beijing, China). The fluorescently labeled probe binds to the template, allowing the *Nfo* enzyme to recognize and cleave the [THF] sites. Following probe cleavage by the endonuclease, the probe was amplified together with the biotin-labeled primer, forming a fluorescently labeled amplification product with biotin labels at both ends. Amplification products with FAM labels at the 5′ end and biotin labels at the 3′ end were formed. These products can be detected using a universal nucleic acid test strip (Fig. S2A and B).

The RAA reaction was performed using the RAA-*Nfo* nucleic acid amplification reagent (Hangzhou Zhongce Bio-Sci & Tech Co., Ltd.) according to the manufacturer's instructions. Briefly, the 50-$\mu$L reaction mixture consisted of 6.5 mg reaction dry powder, 25 $\mu$L buffer A (20% PEG), 2 $\mu$L forward primer (2 $\mu$M), 2 $\mu$L reverse primer (2 $\mu$M), 0.6 $\mu$L probe (2 $\mu$M), 15.9 $\mu$L distilled water, and 2 $\mu$L of the template. Subsequently, 2.5 $\mu$L of buffer B (280 mM Mg(OAc)$_2$) was added to the reaction mixture. After instantaneous centrifugation (4,000 $\times$ *g* for 5 s), the reaction mixture was incubated at 39℃ for 20 min.

The final amplification product carries two labeled groups: the FAM of the probe and the biotin of the downstream primer. Colloidal gold on the test strip and the detection line were labeled with streptavidin and antifluorescein antibodies, respectively. Accordingly, the amplified target products were captured by the test strip, through the binding of their FAM and biotin groups. When labeled target products are present in the sample, they bind to the antibody-labeled gold. Subsequently, they bind to the antibody on the nitrocellulose membrane to form a complex of the antibody-labeled gold and the target detection antibody. Such binding can be identified with the naked eye as a red line, representing a positive result, whereas the absence of a red line indicates a negative result (Fig. S2C). For detection, 50 $\mu$L of the amplification product was diluted with 300 $\mu$L sterile water or phosphate-buffered saline (PBS) solution. Thereafter, the LFS (Hangzhou Zhongce Bio-Sci & Tech Co., Ltd.) sample pad was inserted face down into the diluted reaction solution for 2 min, followed by visual reading of the results.

**Optimization of RAA-LFS detection reaction conditions.** Using the selected forward and reverse primers and probe, $10^3$ fg of genomic DNA per reaction was used as the template for the RAA-LFS detection. The reaction temperature was set to 35 to 45℃ (set as a gradient of 2℃ increments), and the reaction time was set to 5 to 35 min (set as a gradient of 5-min increments). Combined with the results of the test strip, the suitable reaction temperature and time range were screened, and the selected optimal reaction temperature and reaction time were used in subsequent experiments. This experiment was repeated three times.

**Specificity of the RAA-LFS detection technique.** The accuracy of the assay system was evaluated by performing RAA-LFS amplification using 12 *C. auris* reference strains. In addition, to confirm the specificity of the RAA-LFS technique, three strains of *C. pseudohaemulonii*, four strains of *C. duobushaemulonii*, four strains of *C. haemulonii*, nine other common closely related species, and four strains of other pathogenic bacteria were also tested. A total of 12 and 24 isolates of *C. auris* and non-*C. auris* were used for specific detection. This experiment was repeated three times.

**Detection limits of the RAA-LFS detection technique.** First, to determine the detection limit of the genomic DNA of *C. auris* using the detection system, it was diluted 10-fold with a final concentration gradient of $10^7$ to $10^0$ fg/reaction. Simultaneously, to determine the detection limit for the number of cells by the detection system for *C. auris*, its cell suspension was diluted 10-fold with a final concentration gradient of $10^5$ to $10^0$ CFU/$\mu$L. To assess whether contamination of other strains interfered with the sensitivity of the detection method, *C. albicans*, which is commonly encountered in clinical practice, was taken as an example. In the above two protocols, 1 ng of *C. albicans* genomic DNA or $10^5$ CFU/$\mu$L of *C. albicans* DNA was added, and the reaction was performed again to determine the detection limit. According to the detection limit results, the single cells and 10 cells of *C. auris* were cut off by laser microdissection (model LMD6000; Leica Microsystems) for RAA-LFS detection, to verify that the detection system can indeed detect 1 CFU. Next, a small amount of *C. auris* was scraped from SDA plates using a disposable sterile inoculation ring and was thinly applied as a coating on a microscope slide that was precoated with polyethylene naphthalate. Laser microdissection and laser pressure ejection using the Leica LMD6000 system (Leica Microsystems) were performed in a laminar flow biosafety cabinet to capture single cells of *C. auris* in tissue sections under a light microscope. Cells were dissected on a slide using a 337-nm pulsed UV laser and collected in a sample tube. As a negative control, adjacent nonfungal samples of similar size were excised from the same slide and treated in parallel. This experiment was repeated three times.

**Evaluation of the RAA-LFS detection technique using clinical samples.** To evaluate potential sample background interference within the system, the most common blood and urine samples in clinical settings were selected for simulation testing. *C. auris* (strain CBS10913) was mixed with a 200-$\mu$L human whole-blood or urine sample that was subjected to 10-fold serial dilutions. The blood was collected by sterile venous blood collection, 10 mL of whole blood was drawn, 15 g/L of EDTA-2Na was added, and the mixture was mixed well and stored at 4℃. This resulted in final concentrations ranging from $10^5$ to $10^0$ CFU/$\mu$L for the RAA-LFS assay. To determine the effect of human DNA on the detection of *C. auris*, a 10-fold serial dilution of *C. auris* genomic DNA was mixed at a 1:10 ratio with DNA extracted from human blood. This resulted in final concentrations ranging from $10^7$ to $10^0$ fg/reaction for the RAA-LFS assay.

To evaluate the screening effectiveness of this detection system in clinically colonized patients, the nasal cavity with the highest *C. auris* colonization was selected for the simulated nasal swab test. The *C. auris* culture (strain CBS10913) was diluted with a 10-fold PBS gradient; thereafter, 1 $\mu$L of the *C. auris* dilution was dipped in a disposable swab (Shenzhen Medico Technology Co., Ltd., Shenzhen, China) scraped through the nasal cavity for RAA-LFS testing, resulting in final concentrations of $10^5$ to $10^0$ CFU/$\mu$L for the RAA-LFS assay. This experiment was repeated three times.

**Statistical analysis.** Forty-eight simulated samples containing *C. auris* (0 to $10^5$ CFU) were prepared by randomly incorporating *C. auris* into 200 $\mu$L of human whole blood. The 48 simulated urine samples were prepared in the same manner as the blood samples. Similarly, 48 simulated clinical nasal swab samples were prepared. A laser microdissection machine was used to randomly cut different numbers of *C. auris* cells, which were placed in different Eppendorf tubes; the cut cells were dipped in a cotton swab scraped through the nostrils. The assay was performed using the RAA-LFS and qPCR methods. qPCR analysis was performed using the *C. auris* probe fluorescent quantitative PCR kit (Shanghai Yu Bo Biotech Co., Ltd., Shanghai, China). After extracting the DNA as described above, the following components were added to each sample reaction tube according to the manufacturer's instructions: buffer I (2$\times$ probe qPCR MagicMix) at 10 $\mu$L, buffer IV (qPCR

probe) at 1 µL, buffer III (qPCR primer mix) at 2 µL, and sample template DNA at 7 µL. Next, one no-template control (NTC) group was included in each reaction, replacing the sample template DNA with 7 µL of ultrapure water. After all components were added, the mixture was centrifuged briefly (4,000 × *g* for 5 s) and the reaction tube was placed in a real-time qPCR machine. The reaction conditions were set as follows: 95℃ for 3 min, 95℃ for 15 s, and 60℃ for 1 min for 40 cycles. The fluorescence signal of the FAM (5-carboxyfluorescein) channel was collected. A positive result was obtained when the threshold cycle ($C_T$) value was less than 35, whereas the result was negative when it was greater than 35. The kappa conformance test, with positive percent agreement and negative percent agreement, was used to calculate the agreement between the two assays testing the same sample. Statistical analysis was conducted using SPSS software. *P* values of <0.05 were considered significant. Kappa values of ≥0.75 were considered to indicate consistency; kappa values of >0.4 and <0.75 indicated medium consistency, and kappa values of ≤0.4 indicated poor consistency.

**Data availability.** The ITS gene sequences used in this study are available in GenBank *Candida auris*, NCBI reference sequence NR_154998.1; *Candida haemulonii*, reference sequence NR_130669.1; *Candida tropicalis*, reference sequence NR_111250.1; *Candida albicans*, reference sequence NR_125332.1; *Cryptococcus neoformans*, reference sequence NR_171785.1; *Candida parapsilosis*, reference sequence NR_130673.1; *Candida glabrata*, reference sequence NR_130691.1; *Candida krusei*, reference sequence NR_131315.1; *Cryptococcus gattii*, reference sequence NR_165941.1; *Candida duobushaemulonii*, reference sequence NR_130694.1; *Candida pseudohaemulonii*, reference sequence NR_163771.1; *Candida rugosa*, reference sequence NR_111249.1; *Candida guilliermondii*, reference sequence NR_111247.1; *Candida dubliniensis*, reference sequence NR_119386.1.

## SUPPLEMENTAL MATERIAL

Supplemental material is available online only.
**SUPPLEMENTAL FILE 1**, PDF file, 1.2 MB.

## ACKNOWLEDGMENTS

This research was supported by the Major Infectious Diseases such as AIDS and Viral Hepatitis Prevention, the Control Technology Major Projects (2018ZX10101003), and the Consulting Research Project of the Chinese Academy of Engineering (2020-XY-61-03).

All authors made substantial contributions to the work reported, including the study conception, design, and execution, as well as data acquisition, analysis, and interpretation. All authors were involved in preparing, revising, or critically reviewing the manuscript. The order of authors in the byline was determined by seniority. The authors approved the final version for publication and agreed on the journal to which it should be submitted. All authors agree to be accountable for all aspects of the work.

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
