## [Reviewer comments · Microbiology Spectrum]

Microbiology Spectrum

Development of a Novel Method for the Clinical Visualization and Rapid Identification of Multidrug-resistant *Candida auris*

Xin Ran Zhang, Teng Ma, Yu Chen Wang, Shan Hu, and Ying Yang

Corresponding Author(s): Ying Yang, Beijing Institute of Microbiology and Epidemiology

Review Timeline:

Submission Date:	November 29, 2022
Editorial Decision:	December 21, 2022
Revision Received:	February 1, 2023
Editorial Decision:	February 16, 2023
Revision Received:	March 15, 2023
Accepted:	March 25, 2023

Editor: Renato Kovacs

Reviewer(s): Disclosure of reviewer identity is with reference to reviewer comments included in decision letter(s). The following individuals involved in review of your submission have agreed to reveal their identity: Erika P Orner (Reviewer #2)

Transaction Report:

DOI: <https://doi.org/10.1128/spectrum.04912-22>

December 21, 2022

Prof. Ying Yang
Beijing Institute of Microbiology and Epidemiology
Beijing
China

Re: Spectrum04912-22 (Development of a Novel Method for the Clinical Visualization and Rapid Identification of Multidrug-resistant *Candida auris*)

Dear Prof. Ying Yang:

Link Not Available

Sincerely,

Renato Kovacs

Journals Department
Reviewer comments:

Reviewer #1 (Comments for the Author):

Manuscript (Spectrum04912-22) entitled "Development of a Novel Method for the Clinical Visualization and Rapid Identification of Multidrug-resistant *Candida auris*" aimed to develop a rapid and effective method for *C. auris* detection based on recombinase-aided amplification with a lateral flow strip (RAA-LFS).

Although the manuscript is interesting, the table and figures are perfect and the methodology is excellent, it requires major revisions with gross English editing and confirmation or proven for my questions.

The English language requires extensive editing.

Major Comments:

It is not clear what is the clinical significance of the study - how does it help in clinical practice? The major concern is that the

impact of work is not presented enough in the manuscript - authors should at least speculate what could be the importance of their findings in practical application.

Please add significant data about which method was used recently for diagnosing bloodstream *Candida* infections.

Diagnostic methods have a positive impact on clinical outcomes in patients with Candidiasis.

I feel that it is important to mention the efforts to find new combinational methods against these infections

Line 45 authors mentioned " Since its first report in 2009 from Japan, *C. auris* has been increasingly reported in cases of invasive candidiasis (IC).

WGS analyses of clinical *C. auris* isolates indicated that the emergence of clonal populations in Asia, South Africa, and South America occurred independently and spread locally within each region. Based on a newly published paper regarding interesting cases of *auris* from Iran with the literature of review. An isolate representative of a potential fifth clade, distinct from the other clades by >200,000 single-nucleotide polymorphisms, in a patient in Iran who had never traveled outside the country. Please add a paragraph about the prevalence and epidemiological data of *Candida auris* candidiasis. Therefore, those newly published references have been missed; I suggest adding the recent articles both in the introduction and discussion part as follows.

Safari F, Madani M, Badali H, Kargoshaie AA, Fakhim H, Kheirollahi M, Meis JF, Mirhendi H. A Chronic Autochthonous Fifth Clade Case of *Candida auris* Otolomycosis in Iran. *Mycopathologia*. 2022 Feb;187(1):121-127. doi: 10.1007/s11046-021-00605-6. Epub 2021 Dec 2. PMID: 34855102.

Spruijtenburg B, Badali H, Abastabar M, Mirhendi H, Khodavaisy S, Sharifisooraki J, Taghizadeh Armaki M, de Groot T, Meis JF. Confirmation of fifth *Candida auris* clade by whole genome sequencing. *Emerg Microbes Infect*. 2022 Dec;11(1):2405-2411. doi: 10.1080/22221751.2022.2125349. PMID: 36154919; PMCID: PMC9586689.

So, did you check isolates from different clades?

The nomenclature of species should be revised. The first time that the name of each species is mentioned the genus should be written in cursive and then it should be abbreviated

Check reference format and italic form according to the journal format

Clarify your isolates (how many clinical *auris* and non- *C. auris* yeasts)

The specificity and sensitivity test set could be significantly improved by the inclusion of clinically important *auris* species. All these experiments could be presented preceding the testing of clinical samples.

The conclusion in the abstract part was unclear if your results suggested that this assay is suitable for the rapid, sensitive, and specific detection of *C. auris* in clinical samples.

The following reference, you probably missed

This method had high specificity and high sensitivity, which could successfully detect *C. auris* strains.

Please change strains to isolate

This is the limitation of this study. Why you used only *C. auris* isolates, how about other species complex

What are the limitations of this study?

Reviewer #2 (Comments for the Author):

"Development of a Novel Method for the Clinical Visualization and Rapid Identification of Multidrug-resistant *Candida auris*" is a manuscript that describes a novel LFA that can be used to detect *C. auris* DNA after a chelex 100 extraction. While the assay has a lot of potential for clinical use, I believe a few extra experiments should be done to fully elucidate the diagnostic potential of the LFA. See below for specific comments and requests for modification.

General Comments:

Numerous figures were not included in the submission. Tables S1 and S2 and Supplementary Figure 6 were all mentioned but not included. Please submit them for review.

Since this assay requires a boiling step and phenol, I don't believe it would be suitable as a bed-side or POC test, but I do think it has potential for the clinical lab. I would just be careful on how you sell the utility of this test.

Were any of these experiments done in duplicate or triplicate? I believe they should all be replicated to account for variability with extraction or reader bias of the LFA lines.

Is there any way to get true clinical specimens to run on this LFA? It would be good to correlate these to gold standard culture results.

Hospitals are now doing screening tests for *C. auris* colonization by evaluating swab samples (swabbing nose, axilla, and groin). The CDC outlines a protocol for surveillance testing. Though blood infection is important, I would argue surveillance/screening testing for *C. auris* would have more of a clinical impact. I would consider assessing these types of clinical samples as well on your LFA since utilizing this test as proposed would truthfully just be used as a secondary/confirmatory diagnostic for a very small amount of specimens.

Materials and methods section:

-Please elaborate on where the strains were procured from and how they were stored.

-Please include a table of all primers and probes used. I assume this was table S2 but am unsure since it was not included. Also

be sure to reference this table in the text.

-Why was the amplification product diluted?

-Readers are likely more familiar with picograms and larger quantities than fg. I would consider using these measurements instead. I would also consider reporting concentrations based on the input amounts rather than reaction amounts since you are electing to dilute amplification products. Readers will better understand an input CFU to correlate it with clinical relevance. You can always correlate the input CFU to DNA/reaction as well, but I think it's helpful to have the input amount established.

-For the optimization of temperature and timing, it would be better to use your established limit of detection to be sure these parameters work at lower levels of DNA concentrations

-What concentration of organism did you use for the specificity study? It would strengthen the paper if you assess co-culture with *C. auris* and each of the other organisms, not just *C. albicans*.

-Could you reword section 2.8 to reflect you did 2 separate studies, one with CFU and one with extracted DNA, and not that you did one study where you assessed both.

-What does "treated culture" mean? Diluted?

-For the LOD studies, I would like to see lower concentrations since you did not find a concentration it did not detect. Because of this, you cannot conclude a limit of detection. 1 CFU/ μ L is not that low, clinically. It would be good to go down to 1 CFU/mL instead since this is a concentration that can be seen in blood infections.

-How did you obtain whole blood? Define how was it collected and stored.

-Add section that defines the qPCR method you mention. Be sure to include criteria you used to result tests as detected versus not detected.

Results section:

-It is unclear why you chose 37C instead of 39 since the positive line shows up first at that temperature. Just be sure to expand your reasoning.

-Expand on why you chose a 1:10 ratio in section 3.5. Also in this section, I would note there is a slight difference in the strength of positive at 101 between the blood and non-blood samples (blood is slightly lighter).

-Expand section 3.6. Specify which samples were positive and negative by either assay. Also include the qPCR data.

Minor edits:

-Line 43: should read "Candida auris IS a recently discovered...."

-Line 73: I think you meant fungi or microbe, not bacteria

-Paragraph between lines 154-161: Add reference to Supplementary Figure S2A & B

-Line 177: Specifically reference Supplementary Figure S2C

-Line 214: Define the 48 samples

-Line 249: "strength" may be a better term than "depth"

-Line 266: "observed" instead of "detected"

-Figure 4- Specify this is *C. auris* somewhere in the figure caption

-Figure S2 - Include label for biotin in the legend

Staff Comments:

Preparing Revision Guidelines

Please return the manuscript within 60 days; if you cannot complete the modification within this time period, please contact me. If you do not wish to modify the manuscript and prefer to submit it to another journal, please notify me of your decision immediately so that the manuscript may be formally withdrawn from consideration by Microbiology Spectrum.

“Development of a Novel Method for the Clinical Visualization and Rapid Identification of Multidrug-resistant *Candida auris*” is a manuscript that describes a novel LFA that can be used to detect *C. auris* DNA after a chelex 100 extraction. While the assay has a lot of potential for clinical use, I believe a few extra experiments should be done to fully elucidate the diagnostic potential of the LFA. See below for specific comments and requests for modification.

General Comments:

Numerous figures were not included in the submission. Tables S1 and S2 and Supplementary Figure 6 were all mentioned but not included. Please submit them for review.

Since this assay requires a boiling step and phenol, I don't believe it would be suitable as a bed-side or POC test, but I do think it has potential for the clinical lab. I would just be careful on how you sell the utility of this test.

Were any of these experiments done in duplicate or triplicate? I believe they should all be replicated to account for variability with extraction or reader bias of the LFA lines.

Is there any way to get true clinical specimens to run on this LFA? It would be good to correlate these to gold standard culture results.

Hospitals are now doing screening tests for *C. auris* colonization by evaluating swab samples (swabbing nose, axilla, and groin). The CDC outlines a protocol for surveillance testing. Though blood infection is important, I would argue surveillance/screening testing for *C. auris* would have more of a clinical impact. I would consider assessing these types of clinical samples as well on your LFA since utilizing this test as proposed would truthfully just be used as a secondary/confirmatory diagnostic for a very small amount of specimens.

Materials and methods section:

- Please elaborate on where the strains were procured from and how they were stored.
- Please include a table of all primers and probes used. I assume this was table S2 but am unsure since it was not included. Also be sure to reference this table in the text.
- Why was the amplification product diluted?
- Readers are likely more familiar with picograms and larger quantities than fg. I would consider using these measurements instead. I would also consider reporting concentrations based on the input amounts rather than reaction amounts since you are electing to dilute amplification products. Readers will better understand an input CFU to correlate it with clinical relevance. You can always correlate the input CFU to DNA/reaction as well, but I think it's helpful to have the input amount established.
- For the optimization of temperature and timing, it would be better to use your established limit of detection to be sure these parameters work at lower levels of DNA concentrations
- What concentration of organism did you use for the specificity study? It would strengthen the paper if you assess co-culture with *C. auris* and each of the other organisms, not just *C. albicans*.
- Could you reword section 2.8 to reflect you did 2 separate studies, one with CFU and one with extracted DNA, and not that you did one study where you assessed both.
- What does “treated culture” mean? Diluted?

-For the LOD studies, I would like to see lower concentrations since you did not find a concentration it did not detect. Because of this, you cannot conclude a limit of detection. 1 CFU/ μ L is not that low, clinically. It would be good to go down to 1 CFU/mL instead since this is a concentration that can be seen in blood infections.

-How did you obtain whole blood? Define how was it collected and stored.

-Add section that defines the qPCR method you mention. Be sure to include criteria you used to result tests as detected versus not detected.

Results section:

-It is unclear why you chose 37C instead of 39 since the positive line shows up first at that temperature. Just be sure to expand your reasoning.

-Expand on why you chose a 1:10 ratio in section 3.5. Also in this section, I would note there is a slight difference in the strength of positive at 10^1 between the blood and non-blood samples (blood is slightly lighter).

-Expand section 3.6. Specify which samples were positive and negative by either assay. Also include the qPCR data.

Minor edits:

-Line 43: should read "*Candida auris* IS a recently discovered...."

-Line 73: I think you meant fungi or microbe, not bacteria

-Paragraph between lines 154-161: Add reference to Supplementary Figure S2A & B

-Line 177: Specifically reference Supplementary Figure S2C

-Line 214: Define the 48 samples

-Line 249: "strength" may be a better term than "depth"

-Line 266: "observed" instead of "detected"

-Figure 4- Specify this is *C. auris* somewhere in the figure caption

-Figure S2 – Include label for biotin in the legend

Please see our responses to the reviewers' comments below: both the comments and the replies (in blue) are presented.

Reviewer #1 (Comments for the Author):

Manuscript (Spectrum04912-22) entitled "Development of a Novel Method for the Clinical Visualization and Rapid Identification of Multidrug-resistant *C.auris*" aimed to develop a rapid and effective method for *C. auris* detection based on recombinase-aided amplification with a lateral flow strip (RAA-LFS). Although the manuscript is interesting, the table and figures are perfect and the methodology is excellent, it requires major revisions with gross English editing and confirmation or proven for my questions.

1. The English language requires extensive editing.

The language throughout the paper has been carefully checked and improved, and we provide an Editing Certificate as confirmation of this below.

editage

Editing Certificate

This document certifies that the manuscript listed below has been edited to ensure language and grammar accuracy and is error free in these aspects. The logical presentation of ideas and the structure of the paper were also checked during the editing process. Furthermore, a technical review of the manuscript contents was performed. The edit was performed by professional editors at Editage, a division of Cactus Communications. The author's core research ideas were not altered in any way during the editing process. The quality of the edit has been guaranteed, with the assumption that our suggested changes have been accepted and the text has not been further altered without the knowledge of our editors.

MANUSCRIPT TITLE
Development of a Novel Method for the Clinical Visualization and Rapid Identification of Multidrug-resistant Candida auris

AUTHORS
Ying Yang

ISSUED ON
January 14, 2023

JOB CODE
MJIFN_1_17

Vikas Narang
 Chief Operating Officer - Editage

editage

editage, a brand of Cactus Communications, offers professional English language editing and publication support services to authors engaged in over 1300 areas of research. Through its community of experienced editors, which includes doctors, engineers, published scientists, and researchers with peer review experience, Editage has successfully helped authors get published in internationally reputed journals. Authors who work with Editage are guaranteed excellent language quality and timely delivery.

GLOBAL :
+1(833) 979-0061 | request@editage.com

CHINA :
400-120-3020 | 021-6020-9400 |
fabiao@editage.cn

CACTUS

Impact science

researcher life

thejournalsofeditage.com

editage.com | editage.cn | editage.in | editage.org | editage.com.br | editage.com.mx | editage.com.au

2. It is not clear what is the clinical significance of the study - how does it help in clinical practice? The major concern is that the impact of work is not presented enough in the manuscript - authors should at least speculate what could be the importance of their findings in practical application.

For clarity and in accordance with the reviewer's concerns, we have added a short

description in the Discussion and Conclusions as follows: “Pneumonia has been gradually attracting the attention of researchers due to the COVID-19 pandemic, and *Candida* is also an important cause of pneumonia. COVID-19 complicated with *C. auris* infection is a more serious disease, particularly because *C. auris* readily causes nosocomial infections. In many cases, infections caused by *C. auris* are not recognized in the first instance, leading to incorrect medication and delayed treatment [32-36]. (Lines 366 to 371.)

In addition, “The detection method developed in this study allows patients to self-diagnose early. The clinical test results can then be verified and reasonable antifungal treatment promptly initiated.”(Lines 447 to 449.)

3. Please add significant data about which method was used recently for diagnosing bloodstream *Candida* infections.

We have added information about the latest diagnostic methods for *Candida* bloodstream infections in the Discussion section. (Lines 371 to 392.)

4. Diagnostic methods have a positive impact on clinical outcomes in patients with *Candidiasis*.

The original intention of this study was to develop a method for the simple and rapid detection of *C. auris*. Areas that have sub-optimal health facilities do not have a strong awareness of fungal infections, and the increased numbers of patients in such facilities during the COVID-19 pandemic increased the risk of co-infection. The aim of our study was to develop a home testing diagnostic method to enable patients to be diagnosed swiftly and to then promptly seek medical treatment, thereby avoiding delay and further deterioration of the disease due to misdiagnosis or missed diagnosis. Therefore, this method potentially saves lives. In addition, the method enables patients with *Candida* infection to be regularly monitored, which helps to assess disease progression.

5. I feel that it is important to mention the efforts to find new combinational methods against these infections

We agree with you, and we have added information about RAA in the discussion section. RAA is a relatively novel detection technique that was originally applied to detect pathogenic bacteria in viruses or foods. Given its quick results, simple operation, low cost, and the ability to visualize the results, we considered its application in difficult-to-detect pathogenic fungi. (Lines 415 to 418.)

6. Line 45 authors mentioned" Since its first report in 2009 from Japan, *C.auris* has been increasingly reported in cases of invasive candidiasis (IC).WGS analyses of clinical *C.auris* isolates indicated that the emergence of clonal populations in Asia, South Africa, and South America occurred independently and spread locally within each region. Based on a newly published paper regarding interesting cases of auris from Iran with the literature of review. An isolate representative of a potential fifth clade, distinct from the other clades

by >200,000 single-nucleotide polymorphisms, in a patient in Iran who had never traveled outside the country. Please add a paragraph about the prevalence and epidemiological data of *C.auris* candidiasis. Therefore, those newly published references have been missed; I suggest adding the recent articles both in the introduction and discussion part as follows.

Safari F, Madani M, Badali H, Kargoshaie AA, Fakhim H, Kheirollahi M, Meis JF, Mirhendi H. A Chronic Autochthonous Fifth Clade Case of *C.auris* Otomycosis in Iran. *Mycopathologia*. 2022 Feb;187(1):121-127. doi: 10.1007/s11046-021-00605-6. Epub 2021 Dec 2. PMID: 34855102.

Spruijtenburg B, Badali H, Abastabar M, Mirhendi H, Khodavaisy S, Sharifisooraki J, Taghizadeh Armaki M, de Groot T, Meis JF. Confirmation of fifth *C.auris* clade by whole genome sequencing. *Emerg Microbes Infect*. 2022 Dec;11(1):2405-2411. doi: 10.1080/22221751.2022.2125349. PMID: 36154919; PMCID: PMC9586689.

We have now included data about the prevalence and epidemiology of *C. auris* in the Introduction and Discussion sections, and we have also cited the references that you kindly provided. (Lines 52 to 60.)

7. So, did you check isolates from different clades?

The Supplementary Material was unfortunately not fully uploaded, due to an error. The twelve *C. auris* strains used in this study were from four different clades, and specific information about the strains is provided in Table S1 in the Supplementary Material. Our assay was based on the conserved sequence of *C. auris* across all clades, and it is thus reliable and stable for identifying different strains of the same species.

8. The nomenclature of species should be revised. The first time that the name of each species is mentioned the genus should be written in cursive and then it should be abbreviated

Check reference format and italic form according to the journal format

We apologize for our poor formatting and oversight; the full text has now been revised and re-formatted.

9. Clarify your isolates (how many clinical auris and non- *C. auris* yeasts).The specificity and sensitivity test set could be significantly improved by the inclusion of clinically important auris species. All these experiments could be presented preceding the testing of clinical samples.

In this experiment, 12 standard strains of *C. auris* were used to verify the detection stability of this method among the same species. A total of 24 strains of non-auricular *Candida* were also used for specific detection (Lines 219 to 220). Although no more clinical strains were used for further verification in this experiment, our detection method was based on the conserved nucleic acid sequence across the clades of *C. auris*. The experimental results show that our detection method has excellent stability between different strains of the same species and is very specific for the detection of *C. auris*. Due to the impact of the COVID-19 pandemic, it was difficult to collect

clinical strains. However, we hope to apply our detection method to actual clinical samples and compare it with clinical gold standards to evaluate the practical application value of this method in clinical settings. In a follow-up study, we will collect clinical samples to further optimize any problems associated with the detection method and ensure that it is optimally suited to clinical settings. If you have any suitable samples, we would be very pleased if you could share those with us so that we can conduct further tests.

10. The conclusion in the abstract part was unclear if your results suggested that this assay is suitable for the rapid, sensitive, and specific detection of *C. auris* in clinical samples.

The following reference, you probably missed

This method had high specificity and high sensitivity, which could successfully detect *C. auris* strains.

We have inserted the following within the conclusion of the Abstract:

“The simple and cost-efficient detection method established in this study exhibited high specificity and sensitivity and successfully detected *C. auris* in simulated clinical blood samples.” (Lines 29 to 31.)

11. Please change strains to isolate. This is the limitation of this study. Why you used only *C. auris* isolates, how about other species complex

In the subsequent validation and optimization experiments, all the strains used were mixed to evaluate the sensitivity and specificity of the method in the presence of high concentrations of other strains or their DNA. The experimental results showed that the sensitivity and specificity of the detection method were not affected in a complex environment when there were high concentrations of other strains. Considering the limitation of the length of this article, we chose *C. albicans*, which is more commonly encountered in clinical practice, as the representative for quantification. (Please see Lines 425 to 430 in the main manuscript.)

Figure S6 of the Supplementary Material shows the sensitivity and specificity of this detection method in the presence of a high concentration of *C. albicans* bacterial solution and a high DNA concentration.

12. What are the limitations of this study?

We believe that the main limitation of this study is that compared to other swab samples that are easier to handle, the processing of simulated clinical blood samples requires an additional phenol purification step. Therefore, it does not meet the requirements of POCT to the point where it can be used at home or in the field as a commercial product. However, in subsequent experiments, we will consider using fungal nucleic acid extraction reagents that do not require boiling or purification, and which only require blood to be collected from the fingertip or the use of nasal and throat swab and other soaks for nucleic acid extraction. By holding the reaction tube in your hand, subsequent reactions can be activated at body temperature, and the results of disposable nucleic acid test strips can be visualized quickly and easily.

Another limitation of this study is that we unfortunately did not use any real clinical samples, but we selected the most complex blood as a representative of simulated samples, and this enabled us to prove that the detection method can directly identify *C. auris* from blood without affecting the sensitivity. However, we plan to collect actual clinical samples to enable us to further optimize the detection method.

Reviewer #2 (Comments for the Author):

"Development of a Novel Method for the Clinical Visualization and Rapid Identification of Multidrug-resistant *C.auris*" is a manuscript that describes a novel LFA that can be used to detect *C. auris* DNA after a chelex 100 extraction. While the assay has a lot of potential for clinical use, I believe a few extra experiments should be done to fully elucidate the diagnostic potential of the LFA. See below for specific comments and requests for modification.

General Comments:

1. Numerous figures were not included in the submission. Tables S1 and S2 and Supplementary Figure 6 were all mentioned but not included. Please submit them for review.

We apologize for the error in uploading the material correctly. We have now re-uploaded the relevant material and we would be very grateful if you could now review it.

2. Since this assay requires a boiling step and phenol, I don't believe it would be suitable as a bed-side or POC test, but I do think it has potential for the clinical lab. I would just be careful on how you sell the utility of this test.

In this study, the boiling method was used to extract fungal nucleic acids simply and efficiently, while phenol was further used to purify complex blood samples. We tried using swabs, urine, and pure serum samples, all of which only needed to be boiled and did not require further phenol purification. Compared to fungal nucleic acid extraction and purification kits that have complex steps, boiling may be the easiest step to implement at home. In the subsequent optimization of this method, we will use a special fungal nucleic acid simple extraction reagent to achieve POCT (similar to the COVID-19 antigen kit), where a nasal swab or fingertip blood is added to the reagent, and it is then shaken to achieve nucleic acid extraction from *C. auris* without boiling or purification. We anticipate that this will facilitate translation of the test method presented in this study to the field or bedside.

3. Were any of these experiments done in duplicate or triplicate? I believe they should all be replicated to account for variability with extraction or reader bias of the LFA lines.

Each of the experiments in this study was repeated at least three times. We set a large number of replicates with the lowest at 1 CFU, and the results showed a result error rate of only 1% due to strip batches and quality control.

4. Is there any way to get true clinical specimens to run on this LFA? It would be good to correlate these to gold standard culture results.

Due to the impact of Covid-19, it was difficult to collect clinical samples. However, to enable the practical application of this detection method in clinic, we intend to apply it to actual clinical samples, compare it with clinical gold standards, and conduct a subsequent in-depth evaluation of its performance. In follow-up research, we will

collect clinical samples to further optimize problems associated with this detection method. If you could possibly provide us with suitable samples, we would be extremely grateful.

5. Hospitals are now doing screening tests for *C. auris* colonization by evaluating swab samples (swabbing nose, axila, and groin). The CDC outlines a protocol for surveillance testing. Though blood infection is important, I would argue surveillance/screening testing for *C. auris* would have more of a clinical impact. I would consider assessing these types of clinical samples as well on your LFA since utilizing this test as proposed would truthfully just be used as a secondary/confirmatory diagnostic for a very small amount of specimens.

We agree with your comment. It would be best if there were clinical samples that could be verified using our diagnostic methods. We thank you for your suggestion and look forward to the opportunity of working with you to optimize our approach and meet your clinical needs.

Materials and methods section:

6. Please elaborate on where the strains were procured from and how they were stored.

We apologize for our negligence that resulted in the incomplete loading of essential material. All of the supplementary material has now been uploaded and the source information for all strains is provided in Table S1. We have also added the following to Section 2.1,

“All strains were stored in the laboratory at $-20\text{ }^{\circ}\text{C}$ based on the filter paper preservation method.” (Page 6, Lines 111 to 112.)

7. Please include a table of all primers and probes used. I assume this was table S2 but am unsure since it was not included. Also be sure to reference this table in the text.

We apologize for the problem related to the incomplete uploading of materials. All of the primer probe sequences are listed in Table S2, and we would be extremely grateful if you could take the time to now review this.

8. Why was the amplification product diluted?

Due to the HOOK effect of the colloidal gold test strip, false negatives or false positives occur when the ratio of antigens and antibodies participating in the reaction is unsuitable. After amplification, the concentration of antibody-labeled products is high, and if the strip is not diluted, false-negative results are likely (i.e., it is possible that the sample has a very high amount of *C. auris* but the test strip results are negative).

9. Readers are likely more familiar with picograms and larger quantities than fg. I would consider using these measurements instead.

I would also consider reporting concentrations based on the input amounts rather than reaction amounts since you are electing to dilute amplification products. Readers will better understand an input CFU to correlate it with clinical relevance. You can always correlate the input CFU to DNA/reaction as well, but I think it's helpful to have the input amount established.

The common unit of DNA is indeed ng or pg, and if fg was replaced with ng or pg, our detection limit concentration would be expressed as " 10^{-5} ng" or " 10^{-2} pg", which would make illustrations complex. We agree that stating the number of bacteria input, as you propose, is very important, and we apologize that we did not mention that the volume of the fungi suspension at different concentrations was actually 1 μ L. Similarly, 1 CFU/ μ L actually contained 1 CFU of fungi, and we have added this information to Section 2.2. In similar references, the unit of DNA is fg, and the bacterial solution is CFU/ μ L, so we also used this unit.(Lines 127 to 128.)

The associated references are as follows:

1. Wang, L.; Wang, Y.; Wang, F.; Zhao, M.; Gao, X.; Chen, H.; Li, N.; Zhu, Q.; Liu, L.; Zhu, W.; Liu, X.; Chen, Y.; Zhou, P.; Lu, Y.; Wang, K.; Zhao, W.; Liang, W. Development and Application of Rapid Clinical Visualization Molecular Diagnostic Technology for *Cryptococcus neoformans/C. gattii* Based on Recombinase Polymerase Amplification Combined With a Lateral Flow Strip. *Front. Cell. Infect. Microbiol.* **2021**, *11*, 803798.
2. Wang, F.; Ge, D.; Wang, L.; Li, N.; Chen, H.; Zhang, Z.; Zhu, W.; Wang, S.; Liang, W. Rapid and sensitive recombinase polymerase amplification combined with lateral flow strips for detecting *Candida albicans*. *Anal. Biochem.* **2021**, *633*, 114428.

10. For the optimization of temperature and timing, it would be better to use your established limit of detection to be sure these parameters work at lower levels of DNA concentrations

The results in Figure S5 show that the detection limit of this method is 10^1 fg/reaction. However, the bands of 10^1 and 10^2 fg/reaction are faint, and if the reaction condition optimization is performed with these two template concentrations, the reaction time needs to be greatly extended and the reaction temperature increased to normalize the brightness of the resulting bands. To increase the convenience and speed of using the detection method at body temperature, which would be more in line with the POCT conditions, we selected 10^3 fg/reaction as the template concentration for optimizing the reaction conditions.

11. What concentration of organism did you use for the specificity study? It would strengthen the paper if you assess co-culture with *C. auris* and each of the other organisms, not just *C. albicans*.

In the specificity study, we used 10^5 CFU/mL in 1 μ L for DNA extraction and detection, and this information has now been provided in Section 2.2. (Lines 127 to 128.) We apologize for our error. In the experiment, we mixed all strains at high concentrations to detect *C. auris*, and found that the sensitivity remained unchanged;

however, considering the word limit of the article, we selected the bacteria most commonly encountered in clinical settings, *Candida albicans*, as an example. We then quantified the values and presented the results. We have included information about mixing in the Discussion section to support our results. (Lines 425 to 430.)

12. Could you reword section 2.8 to reflect you did 2 separate studies, one with CFU and one with extracted DNA, and not that you did one study where you assessed both.

We have rewritten Section 2.8 and have separated the two experiments using DNA and CFU to improve the readability and avoid ambiguity. (Lines 223 to 231.)

13. What does "treated culture" mean? Diluted?

"Treated culture" meant that 1 μL of the bacterial solution of *C. albicans* was taken from different concentrations for DNA extraction, as described in Section 2.2. (Lines 122 to 124.)

14. For the LOD studies, I would like to see lower concentrations since you did not find a concentration it did not detect. Because of this, you cannot conclude a limit of detection. 1 CFU/ μL is not that low, clinically. It would be good to go down to 1 CFU/mL instead since this is a concentration that can be seen in blood infections.

Due to our negligence, the actual amount of *C. auris* used was not specified, and our detection limit of "1 CFU/ μL " meant that the sampling volume of 1 μL at a concentration of 1 CFU/ μL could be detected, which implied that 1 *C. auris* could be detected. To verify this result, we used microdissection to cut individual cells for DNA extraction and detection. The experimental results proved that 1 CFU could indeed be detected. The figure below shows the microdissection and test results. (Lines 425 to 430.)

15. How did you obtain whole blood? Define how was it collected and stored.

All the blood samples in this study were collected by the researchers using sterile venous blood collection kits to draw a certain amount of their own whole blood. Subsequently, 15 g/L of EDTA-2Na was added, and the contents were mixed and

stored at 4°C. This information has now been added to Section 2.9. (Lines 235 to 237.)

16. Add section that defines the qPCR method you mention. Be sure to include criteria you used to result tests as detected versus not detected.

The qPCR reaction in Section 2.10 was performed using a commercial kit. “A positive result was obtained when the Ct value was less than 35, whereas the result was negative when it was greater than 35.” This information has now been added to Section 2.10. (Lines 246 to 249.)

Results section:

17. It is unclear why you chose 37°C instead of 39 since the positive line shows up first at that temperature. Just be sure to expand your reasoning.

We believe that although 39 °C is the first temperature to show a positive result, it is not an easily attainable temperature at home or at the bedside. An appropriate extension of the reaction time to 15 min at 37 °C showed stably positive results, and when the time was prolonged and the temperature increased, the positive results were maintained. Considering the ease of implementing the reaction time and temperature, we chose 37 °C and 15 min as the reaction conditions. We have added such information to Section 3.2, where we briefly explain the reasons for selecting 37 °C. (Lines 301 to 303.)

18. Expand on why you chose a 1:10 ratio in section 3.5. Also in this section, I would note there is a slight difference in the strength of positive at 10¹ between the blood and non-blood samples (blood is slightly lighter).

We chose a 1:10 dilution based on the amount used in a similar article that employed simulated blood samples to diagnose *C. auris*. The relevant article is as follows:

1. Sattler, J.; Noster, J.; Brunke, A.; Plum, G.; Wiegel, P.; Kurzai, O.; Meis, J.F.; Hamprecht, A. Comparison of Two Commercially Available qPCR Kits for the Detection of *Candida auris*. *J. Fungi* **2021**, *7*.

The light color of the bands in the blood sample occurs because the complex blood background contains a variety of impurities. However, for the test strip results, the positive band shows that our detection method can maintain a sensitivity of 1 CFU in a complex sample, despite the complexity of the test background.

19. Expand section 3.6. Specify which samples were positive and negative by either assay. Also include the qPCR data.

In order to better simulate detection in unknown samples, the process of mixing *C. auris* into blood samples and the process of detecting samples were completely separated and a double-blind experimental process was used. The qPCR curve is shown below, and detailed Ct values of each sample and the results of the two assays are supplemented in Table S3 in the Supplementary Material. We would be extremely grateful if you could please review this information (Lines 354 and 356.)

Minor edits:

20. Line 43: should read "*C.auris* IS a recently discovered...."

We apologize for our careless mistake and have now corrected this information. (Line 50.)

21. Line 73: I think you meant fungi or microbe, not bacteria

We sincerely thank you for reading our manuscript so carefully. Based on your recommendation, we have corrected "bacteria" to "fungi." (Line 95.)

22. Paragraph between lines 154-161: Add reference to Supplementary Figure S2A &B

We apologize for our careless mistake and have now made associated changes. (Line 185.)

23. Line 177: Specifically reference Supplementary Figure S2C

We apologize for our careless mistake and have now made associated changes. (Line 201.)

24. Line 214: Define the 48 samples

We sincerely thank you for reading our manuscript so carefully. Based on your recommendation, "the 48 samples" are defined as follows, "Forty-eight simulated samples containing *C. auris* ranging from 0 - 10⁵ CFU were prepared by randomly incorporating *C. auris* into 200 µL of human whole blood." (Lines 244 to 245.)

25. Line 249: "strength" may be a better term than "depth"

We sincerely thank you for reading our manuscript so carefully. Based on your recommendation, we have corrected "depth" to "strength." (Line 296.)

26. Line 266: "observed" instead of "detected"

We sincerely thank you for reading our manuscript so carefully. Based on your recommendation, we have changed "detected" to "observed." (Line 313.)

27. Figure 4- Specify this is *C. auris* somewhere in the figure caption

We apologize for our careless mistake and have now revised this. We have changed the legend of Figure 4 to, "Detection of standard reference strains of *C. auris*." (Line 635.)

28. Figure S2 - Include label for biotin in the legend

We again apologize for our careless mistake, and have made associated corrections by adding the label of biotin to the legend of Figure S2.

February 16, 2023

Prof. Ying Yang
Beijing Institute of Microbiology and Epidemiology
Beijing
China

Re: Spectrum04912-22R1 (Development of a Novel Method for the Clinical Visualization and Rapid Identification of Multidrug-resistant *Candida auris*)

Dear Prof. Ying Yang:

Link Not Available

Sincerely,

Renato Kovacs

Journals Department
Editor comments:

As you read below, several concerns remained unanswered or unclarified. Please, provide detailed explanations for Reviewer(s) about the raised problems; furthermore, involve the relevant information into the manuscript focusing on the methodological background and clinical applicability of your test.

Reviewer (Comments for the Author):

I do not feel previous comments have been thoroughly addressed. Though some have been fully addressed, some were only addressed in comments back to the reviewers and not put in the manuscript, some were partially addressed, and some were not addressed at all.

The biggest issue that was not fully addressed is that the authors have still not clarified the clinical impact of this test. As it stands, it cannot be used as a point of care test because it requires a place to set up a nucleic acid amplification which not only requires a tech to use pipettes and a clean space, but it also requires a phenol and boiling step, none of which could be done at

home or at the bedside. Further, this is a nosocomial infection (as even pointed out by the authors) where patients could have a pneumonia (where concentrations in the blood may not be that high and therefore not picked up unless grown in blood culture first). Even if a patient were fungaemic, concentrations of yeast in the blood may still be low and require a positive blood culture. You can look at the literature on the T2 Candida panel and see even with their NAAT test, there's only a certain population of patients where this test is most clinically impactful. Because of all these issues, I believe the authors need to reconsider the utility of this test. There is still utility for this in an under-resourced setting, but it still would only have applicability in a lab setting. I think the authors need to completely rework how they present this test and what clinical impact and utility it would have. The other comments that I think need to be further addressed are the following:

The English still needs work

Any rebuttal the authors gave to the reviewer comments need to be addressed in the manuscript as well. For example, you state that all experiments were done three times, yet this is not stated in the manuscript, nor is there any evidence in the data.

There not only as some methodologies that still are not outlined in the methods section (i.e. qPCR), but with the new edits, you mention a microdissection technique that is not mentioned in the manuscript at all. This not only needs to be added, but you must include specifics in the methodology section. You must also be specific with your methods. You cannot say "generally by a factor of 6" [line 434] or "a certain amount of whole blood" [line 235].

The limit of detection of your assay is not consistently stated throughout the manuscript. In some sections it is 1 CFU, in others it is 2 CFU. Similarly in section 3.6 you say there are 48 simulated samples but you only give results for 19.

Staff Comments:

Preparing Revision Guidelines

Please return the manuscript within 60 days; if you cannot complete the modification within this time period, please contact me. If you do not wish to modify the manuscript and prefer to submit it to another journal, please notify me of your decision immediately so that the manuscript may be formally withdrawn from consideration by Microbiology Spectrum.

Please see our responses to the reviewers' comments below: both the comments and the replies (in blue) are presented.

We sincerely thank the editor and the reviewers for their valuable feedback that has helped us improve the quality of our manuscript. The editor and reviewer comments are presented in bold, and the specific concerns have been numbered. Our response is presented in normal font, and changes/additions to the manuscript are given in blue.

Editor comments:

As you read below, several concerns remained unanswered or unclarified. Please, provide detailed explanations for Reviewer(s) about the raised problems; furthermore, involve the relevant information into the manuscript focusing on the methodological background and clinical applicability of your test.

On behalf of my co-authors, we thank you for giving us a chance to revise and improve the quality of our article. We have read the reviewers' and your comments carefully and have made revision which marked in blue in the paper. We have tried our best to revise our manuscript according to the comments. Attached please find the revised version, which we would like to submit for your kind consideration. Here, we would like to explain the changes briefly as follows:

We have found an English native speaker with a research background to review our manuscript during revision. And if you think there is any problem, you can raise it at any time. we will look for professional organizations to improve the language.

Regarding the need to supplement the methodological background and redefine clinical applicability as raised by the reviewers, we have supplemented the material methodology section of the manuscript and made changes to the clinical suitability description of the full text.

We have written a point-by-point response letter for reviewers, you can see the details at the end of this letter. To make the reply more visible, Q represents questions raised by reviewers, and A are our answers for these questions. In all, we found these comments are quite helpful. And special thanks to you and reviewers for your good comments again.

I wish this revision will be acceptable for publication in your journal.
Thank you for your consideration. I am looking forward to hearing from you.

Reviewer (Comments for the Author):

Q1: I do not feel previous comments have been thoroughly addressed. Though some have been fully addressed, some were only addressed in comments back to

the reviewers and not put in the manuscript, some were partially addressed, and some were not addressed at all.

A: We are very grateful to your comments for the manuscript. According with your advice, we tried our best to amend the relevant part and made some changes in the manuscript. These changes will not influence the framework of the paper. All of your questions were answered below. And here we list the changes and marked in blue in revised paper.

We appreciate for Reviewers' warm work earnestly, and hope that the correction will meet with approval. Should you have any questions, please contact us without hesitate.

Once again, thank you very much for your comments and suggestions.

Q2: The biggest issue that was not fully addressed is that the authors have still not clarified the clinical impact of this test. As it stands, it cannot be used as a point of care test because it requires a place to set up a nucleic acid amplification which not only requires a tech to use pipettes and a clean space, but it also requires a phenol and boiling step, none of which could be done at home or at the bedside.

Further, this is a nosocomial infection (as even pointed out by the authors) where patients could have a pneumonia (where concentrations in the blood may not be that high and therefore not picked up unless grown in blood culture first). Even if a patient were fungaemic, concentrations of yeast in the blood may still be low and require a positive blood culture. You can look at the literature on the T2 Candida panel and see even with their NAAT test, there's only a certain population of patients where this test is most clinically impactful. Because of all these issues, I believe the authors need to reconsider the utility of this test. There is still utility for this in an under-resourced setting, but it still would only have applicability in a lab setting. I think the authors need to completely rework how they present this test and what clinical impact and utility it would have.

A: Thank you for your valuable comments. Due to the current reaction conditions, the use of pipettes cannot be eliminated. To avoid confusion among readers, we have shifted the potential application of this technology to hospitals in remote or medically underdeveloped areas that require screening for *C. auris* infection. Simultaneously, we have removed the statement that "blood samples can be directly tested without culture." Besides, we selected the blood background as a representative of a complex sample to demonstrate that our detection system is not affected by interference from the complex sample background.

In response to your question about the challenges of detecting *C. auris* bloodstream infections and the reports indicating that the most frequent colonization of patients

with *C. auris* occurs in the nostrils, we have incorporated simulated nasal swab and urine samples. We assessed the detection sensitivity and comparative detection accuracy of these two samples using qPCR. These two sample types were obtained noninvasively and without further purification using phenol-chloroform for DNA extraction. The primary purpose of including these two simulated samples was to verify if our detection system could be used to screen patients colonized by *C. auris*. The corresponding experimental methods are described in sections 2.9 and 2.10, and the experimental results are presented in sections 3.5 and 3.6.

The other comments that I think need to be further addressed are the following:

Q3: The English still needs work

A: Thank you very much for your comments. We have sought the assistance of Dr. Katharina, a well-established expert, to refine our paper. Please let us know if the revised version meets the English presentation standards.

Q4: Any rebuttal the authors gave to the reviewer comments need to be addressed in the manuscript as well. For example, you state that all experiments were done three times, yet this is not stated in the manuscript, nor is there any evidence in the data.

A: Thank you for your valuable suggestions. In response, we have specified that all experiments were conducted in triplicate within each subsection of the Materials and Methods section. Additionally, since our experimental results were qualitative, we carried out a semi-quantitative analysis of the grayscale band intensity for the test line (line 390). The findings are displayed in Fig. S8 of the supplementary materials, and the results of the triplicate experiments are represented by error bars.

Q5: There not only as some methodologies that still are not outlined in the methods section (i.e. qPCR), but with the new edits, you mention a microdissection technique that is not mentioned in the manuscript at all. This not only needs to be added, but you must include specifics in the methodology section. You must also be specific with your methods. You cannot say "generally by a factor of 6" [line 434] or "a certain amount of whole blood" [line 235].

A: Thank you for your valuable comments. We have incorporated a detailed description of the qPCR reaction method in Section 2.10 [line 276] and a comprehensive explanation of the microdissection method in Section 2.8 [line 237].

In response to your feedback, we have revised the phrase "generally by a factor of 6" to "During the experiments, the product obtained in the RAA reaction was diluted 6-fold to avoid false-negative results caused by high product concentration in the test strip" [line 475].

Similarly, we have amended "a certain amount of whole blood" to "10 mL of whole blood was drawn" [line 248].

Q6: The limit of detection of your assay is not consistently stated throughout the manuscript. In some sections it is 1 CFU, in others it is 2 CFU. Similarly in section 3.6 you say there are 48 simulated samples but you only give results for 19.

A: We apologize for any lack of clarity in our previous version. We have now ensured that the description of detection limits throughout the manuscript consistently refers to 1 CFU. In Section 3.6, we discuss the preparation of blood samples, which included 19 (out of 48) positive samples and the remaining 29 negative samples. This is illustrated in the qPCR result plot, which shows 19 positive curves (red lines in the figure) and 29 negative lines (blue lines in the figure). The results for all 48 blood samples can be found in Table S3 of the supplementary material.

We have compiled the results for simulated urine samples and simulated nasal swab samples in a similar manner (line 265), which are presented in Tables S4 and S5 of the supplementary material, respectively. The qPCR raw data plots for these two sets of simulated samples are provided below.

Simulated qPCR raw data of blood samples

Simulated qPCR raw data of urine samples

Simulated qPCR raw data of nasal swab samples

March 25, 2023

Prof. Ying Yang
Beijing Institute of Microbiology and Epidemiology
Beijing
China

Re: Spectrum04912-22R2 (Development of a Novel Method for the Clinical Visualization and Rapid Identification of Multidrug-resistant *Candida auris*)

Dear Prof. Ying Yang:

Your manuscript has been accepted, and I am forwarding it to the ASM Journals Department for publication. You will be notified when your proofs are ready to be viewed.

Sincerely,

Renato Kovacs
Editor, Microbiology Spectrum
